# The Effects of Curcumin Nanoparticles Incorporated into Collagen-Alginate Scaffold on Wound Healing of Skin Tissue in Trauma Patients

**DOI:** 10.3390/polym13244291

**Published:** 2021-12-08

**Authors:** Mohammadmahdi Mobaraki, Davood Bizari, Madjid Soltani, Hadi Khshmohabat, Kaamran Raahemifar, Mehdi Akbarzade Amirdehi

**Affiliations:** 1Biomaterial Group, Faculty of Biomedical Engineering (Center of Excellence), Amirkabir University of Technology, Tehran 15916-34311, Iran; m.mahdimobaraki70@gmail.com; 2Trauma Research Center, Baqiyatallah University of Medical Sciences, Tehran 14351-16471, Iran; khoshmohabat@yahoo.com; 3Department of Mechanical Engineering, K. N. Toosi University of Technology, Tehran 19395-19919, Iran; 4Department of Electrical and Computer Engineering, University of Waterloo, Waterloo, ON N2L 3G1, Canada; 5Centre for Biotechnology and Bioengineering (CBB), University of Waterloo, Waterloo, ON N2L 3G1, Canada; 6Advanced Bioengineering Initiative Center, Multidisciplinary International Complex, K. N. Toosi University of Technology, Tehran 14176-14411, Iran; 7College of Information Sciences and Technology (IST) Data Science and Artificial Intelligence Program, Penn State University, State College, PA 16801, USA; kraahemi@gmail.com; 8Chemical Engineering Department, University of Waterloo, 200 University Avenue West, Waterloo, ON N2L 3G1, Canada; 9Faculty of Science, School of Optometry and Vision Science, University of Waterloo, Waterloo, ON N2L 3G1, Canada; 10MSc Student of Medical–Surgical Nursing, Head of Wound and Ostomy Clinic of Baqiyatallah Hospital, Baqiyatallah University of Medical Science, Tehran 14359-16471, Iran; mehdiakbarzadeh_esmd@yahoo.com

**Keywords:** wound healing, nanocurcumin, cell viability, collagen, alginate, scaffold

## Abstract

Wound healing is a biological process that is mainly crucial for the rehabilitation of injured tissue. The incorporation of curcumin (Cur) into a hydrogel system is used to treat skin wounds in different diseases due to its hydrophobic character. In this study, sodium alginate and collagen, which possess hydrophilic, low toxic, and biocompatible properties, were utilized. Collagen/alginate scaffolds were synthesized, and nanocurcumin was incorporated inside them; their interaction was evaluated by FTIR spectroscopy. Morphological studies investigated structures of the samples studied by FE-SEM. The release profile of curcumin was detected, and the cytotoxic test was determined on the L929 cell line using an MTT assay. Analysis of tissue wound healing was performed by H&E staining. Nanocurcumin was spherical, its average particle size was 45 nm, and the structure of COL/ALG scaffold was visible. The cell viability of samples was recorded in cells after 24 h incubation. Results of in vivo wound healing remarkably showed CUR-COL/ALG scaffold at about 90% (*p* < 0.001), which is better than that of COL/ALG, 80% (*p* < 0.001), and the control 73.4% (*p* < 0.01) groups at 14 days/ The results of the samples’ FTIR indicated that nanocurcumin was well-entrapped into the scaffold, which led to improving the wound-healing process. Our results revealed the potential of nanocurcumin incorporated in COL/ALG scaffolds for the wound healing of skin tissue in trauma patients.

## 1. Introduction

Curcumin is a hydrophobic agent that is mainly extracted from turmeric rhizome, and exhibits wound-healing, antibacterial, anti-inflammation, antioxidant, and anticancer activities [1,2,3,4,5]. Regarding curcumi characteristics n, recent reports showed that it can be effective against many types of cancer, such as breast and prostate cancer [6]. Curcumin is a hydrophobic polyphenol that has many limitations such as its low solubility in water and rapid metabolism. To overcome these obstacles, the synthesis of curcumin nanoparticles was considered since they can prolong circulation and permeability [7,8]. Polyethylene glycol (PEG) is hydrophilic and attractive candidate for therapeutic applications, so it generally improves solubility in hydrophilic solvents and the stability of nanoparticles [9,10]. During the past two decades, collagen has received increased attention for fabricating biomaterials for tissue regeneration with unique properties, such as outstanding biocompatibility, hydrophilicity, and biodegradability [11,12].

Additionally, hydrogels are utilized for cell scaffolds for promoting tissue formation and enhancing cell proliferation [13], such as for skin damage caused by different events and diseases, such as burns, metabolic diseases, infections, trauma, and tumors. Evidence shows that wounds associated with trauma or surgery are extended in about 300 million people globally [14]. CUR has drawn attention towards skin wound healing treatment due to some of its properties, including low toxicity and pharmacological properties, and its distinguishing features make it a good factor for wound healing [15,16]. Alginate–gelatin microfibers constructed with curcumin have great potential for wound healing [17]. The majority of papers provided evidence that curcumin with natural polymers such as alginate, chitosan, collagen, and gelatin improved wound healing in a variety of diseases [17,18,19]. Sodium alginate (SA) is an anionic polysaccharide [20], which is one of the most popular anionic polymers utilized in biomedical applications due to its unique features such as hydrophilicity, biodegradability, and nontoxicity, and it plays a crucial role in the drug delivery process [9,21,22]. This paper utilizes nanocurcumin for improving problems of curcumin such as its poor bioavailability and low stability. Nanocurcumin was incorporated into a COL/ALG scaffold for evaluating wound healing in in vitro and in vivo studies in diseases related to trauma. We used PEG in this work to PEGylate nanocurcumin, which can be effective for reducing immune response. We evaluate the characterization, release, and cell viability of curcumin nanoparticles and scaffolds.

## 2. Materials and Methods 

### 2.1. Materials

Fetal bovine serum (FBS), Dulbecco’s phosphate buffered saline (PBS), Dulbecco’s modified Eagle’s medium (DMEM), penicillin/streptomycin, and trypsin were obtained from Gibco (Carlsbad, CA, USA), Fetal bovine serum (FBS) and curcumin (CUR) were obtained from Sigma-Aldrich, Australia. Thiazolyl Blue Tetrazolium Bromide (MTT), dimethylsulfoxide (DMSO), PEG, alginate (ALG), and collagen (COL) were purchased from Sigma (Sigma–Aldrich, St. Louis, MO, USA). All other materials utilized in this study were from domestic providers at analytical grade.

### 2.2. Preparation of Collagen/Alginate Scaffolds Impregnated with Curcumin-PEG Nanoparticles (CUR-PEG nps/ALG/CL)

Collagen solution (1% wt) was prepared by dissolving collagen in 0.5 M acetic acid and alginate powder solved in distilled water to achieve an alginate solution (2% wt). Collagen–alginate mixtures were prepared. The pH of acidic collagen was adjusted to 7 by adding 2 M NaOH at 4 °C. The alginate solution was added dropwise to the collagen solution. The mixture was stirred for 2 h to obtain a clear homogeneous blend, which was then centrifuged at 4000 rpm for 15 min for the removal of entrapped air bubbles. The hydrogels were then washed with deionized water collagen–alginate composite scaffolds with a collagen/alginate ratio of 50/50 (*w*/*w*), prepared, and frozen at −40 °C for 24 h, followed by lyophilization to obtain a ALG/COL scaffold. In order to synthesize nano-CUR, 10 mg of curcumin was dissolved into 10 mL acetone that had been magnetically stirred at 500 rpm for 1 h. Then, the mixture was homogenized in an ultrasonic bath (DT510, Bandelin, 35 kHz, Germany) for 20 min. Additionally, nanoparticles were collected by centrifugation at 16,000× *g* for 30 min. The CUR (PEG6000, 50%) solution was sonicated for 5 min and then vigorously mixed overnight using a magnetic stirrer. Afterwards, mixtures were filtered through a 0.45 µm filter. Then, 1 mL of CUR-PEGNPs 8 mg/mL was added to 10 mL ALG/COL composite for 4 h, followed by dialysis and stirred overnight; scaffolds were freeze-dried and stored in a desiccator [18].

### 2.3. Characterization Methods

Fourier transform infrared spectroscopy (FTIR) was performed using an FTIR spectrometer (Bruker, Bremen, Germany) to understand the interaction between the constituents of the particles. CUR-PEG/ALG/COL, CUR, COL: Samples determined as KBr discs. Morphological structures of the samples were studied by field emission scanning electron microscopy (FE-SEM) (SigmaVP, Carl Zeiss, Oberkochen, Germany) at 20 kV voltage after coating each sample with a thin layer of gold for 5 min. The related size of samples was also estimated using this instrument. Parameters such as the size of the CUR/PEG nanoparticles were established with a Scattering Particle Size Analyzer (Malvern Co., Malvern, UK).

### 2.4. Cell Cultures

L929 cells were cultured with 8 × 10^4^ cell/well at 37 °C at 5% CO_2_ in RPMI-1640 medium supplemented with 1% penicillin/streptomycin solution and 10% fetal bovine serum (FBS). 

### 2.5. MTT Cell Viability Assay

A 3-[4, 5-dimethylthiazol-2-yl]-2, 5-diphenyltetrazolium bromide MTT assay for cell culture was obtained according to the method by Mossman [23]. The assay for the cell cultures was carried out with a standard protocol. They were plated into 96-well culture plates. After 24 days of treatment (concentrations of 15, 25, 35, 45, 55, and 65 of CUR, ALG/COL, CUR/ALG/COL were utilized with 3 replicates), 10 µL of MTT (5 mg mL^−1^) in PBS solution was added into each 96-well plate, and incubated for 4 h at 37 °C, Then, the media were removed from each well and 190 µL of DMSO was added. Absorbance was detected at 570 nm by immuneabsorbent assay (ELISA) (multimode reader, synergy HTX, BioTek, Winooski, VT, USA [24].

### 2.6. Determination of Release Profile

Release of curcumin from nanoparticles (10 mg curcumin dissolved in 10 mL PBS) was measured in phosphate-buffered saline (PBS) at different time intervals. The sample was dialyzed by dialysis membrane (MWCO 12,000 Da) against PBS, and free curcumin in the supernatant was quantified with an Optizen 3220 (UV, South Korea) instrument at 430 nm absorbance [5,16].

### 2.7. Wound Healing, and Hematoxylin and Eosin Staining

All female rats were 6–7 weeks old, weighed 145–170 g, and were anesthetized by injection of 10% chloral hydrate (3 mL/kg body weight). Skin sections with a diameter of 15 mm were established. A total of 21 rats with induced skin sections were divided into 3 groups with 7 rats in each group: (1) blank control group (*n* = 7), (2) COL/ALG (*n* = 7), (3) CUR/COL/ALG (*n* = 7). Samples were intradermally administered with 0.9% *w/v* NaCl, 1 mg COL/ALG, and 1 mg CUR/COL/ALG by sterile insulin syringe (BD medical, France); 100 µL of samples were then applied to the wound surface. Wound changes were investigated in all rats, and the wounds were photographed at 0, 7 and 14 days after surgery. Skin sections were cut and immersed in normal 10% buffered formalin for hematoxylin and eosin (HE) staining to assess the predominant stages of healing.

### 2.8. Statistical Analysis 

Data are shown as mean ± SD. Statistical analysis was performed by one-way ANOVA. Values of *p* < 0.01 *, *p* < 0.001 ** and *p* < 0.0001 *** were indicative of statistically significant differences.

## 3. Results

### 3.1. Physical Characterization of CUR-nps/ALG/COL

#### FTIR-Spectrometer of CL, CUR-nps/ALG/CL, CUR, ALG

As you can see in Figure 1, the broad absorption band at 2902–3334 cm^−1^ corresponds to stretch O-H bond existing in the pure ALG and also the pure alginate spectrum assigned at the peaks at 1028 cm^−1^ is illustrated in Figure 1A; the intensity absorption band at 1089 cm^−1^ was dependent on the C-O-C bond in PEG, indicated in Figure 1D. The spectral peak of free curcumin was demonstrated close to 3509 cm^−1^ (Figure 1C); this peak was assigned to the free hydroxyl group. COL peaks around 2927 cm^−1^ and 2854 cm^−1^ provide evidence for the –CH groups shown in Figure 1B. 

The average size of samples was recorded by dynamic light scattering; empty CUR nanoparticles at room temperature at 68 nm (Figure 2A.) and COL/ALG (Figure 2B.) were investigated by field-emission scanning electron microscopy (FE-SEM), and were uniformly spherical.

### 3.2. Antiproliferative and Cytotoxic Effects of Nano-Spheres In Vitro

Cell were treated for 24 h with empty free curcumin and CUR/COL/ALG, and incubated for another 24 h in fresh culture medium. Cell viability was investigated by MTT assay (Figure 3), which showed CUR/COL/ALG to be better than the free CUR.

### 3.3. Release Profile of Curcumin

The release pattern of curcumin is shown in Figure 4. Data showed that 0, 14.25, 21.2, 26.88, 31.37, 85.22 and 94.87% of the curcumin was released in 0, 1, 2, 3, 4, 16 and 24 h, respectively.

### 3.4. Hematoxylin and Eosin (H&E)

Skin sections of skin layers were stained with hematoxylin and eosin (H&E) during the wound-healing process. Results of the wound tissue with the control, CUR, COL/ALG and CUR/COL/ALG on 7 and 14 days of postwounding are illustrated in Figure 5. The CUR/COL/ALG group presented a remarkably good healing effect on the tissue on the 7th and 14th days. On days 7 and 14, wounds showed scab formation only in the case of the control, the COL/ALG and CUR/COL/ALG treated groups did not show it. Additionally, the epidermal tissue was observed in all groups except the control on the 7th day. Tissue analysis on day 7 demonstrated granulation tissue formation in the control and COL/ALG groups. The CUR/COL/ALG scaffold accelerated wound recovery rate and showed wound re-epithelialization, and growing hair follicles and blood vessels on days 7 and 14.

### 3.5. Wound Healing

The effects of 0.5 mg CUR, 1 mg COL/ALG and 1 mg CUR/COL/ALG on wound healing were investigated in trauma model rates. Wound area was considered during periods of 7 and 14 days (Figure 6). CUR/COL/ALG demonstrated better wound healing. COL/ALG showed clear healing compared to the control. The wound area of CUR/COL/ALG, COL/ALG and control had recovered by about 90% (*p* < 0.001), 80% (*p* < 0.001), and 73.4% (*p* < 0.01) at 14 days, respectively (Figure 6).

## 4. Discussion

Curcumin is poorly absorbed due to its low solubility in water. Studies indicated that chitosan nanoparticle (CN)-incorporated collagen-chitosan scaffolds are vitally important, particularly for promoting wound healing. The water absorption capacity of scaffolds can transport nutrients and metabolic products into the construct [19,25]. Reports proposed the use of CUR-CSNP-loaded ALG/COL scaffolds to improve dermal wound healing by reducing inflammation in diabetic rats. Natural polymers for curcumin impregnated in scaffolds were used here. In previous reports of toxic polymers, they were applied, and problems arose, such as not being able to easily remove the reaction [21]. Recorded results show that CUR alone have problems in clinical applications such as side effects, limit absorption, and fast metabolism [26]. Similar results demonstrated that PEG nanoparticles may inhibit the phagocytosis process. Moreover, PEG into PNPs mainly improved drug circulation in blood and permeability in cancer cells, and these nanoparticles enhanced drug pharmacokinetics in blood [10,27,28]. Studies showed that CUR-PNPs exhibited stronger cytotoxicity than that of the CUR solution in rat glioma cells (Table 1) [10]. 

Other reports showed that curcumin-loaded PLGA NPs coated with chitosan and PEG can significantly enhance apoptosis and cytotoxicity compared to free curcumin [29]. Recent findings indicated that curcumin-loaded CG/Alg nanoparticles are suitable for the delivery of curcumin to carcinoma cells MCF-7 [10,20,29]. Investigated studies revealed that the GC/L/C bilayer nanofibrous scaffolds play a significant role in chronic wound treatment [16]. Results showed that encapsulating cc inside PLGA polymer increases the stability and solubility of CC and accelerates wound-healing processes [30]. It was also found that wounds were treated by the Cur/HA were particularly effective in rehabilitating tissue to normal after 14 days and demonstrated 96% wound healing. Moreover, Hematoxylin and eosin (HE) staining presented that hair follicles in the Cur/HA group were illustrated [15]. Findings indicate that nanoparticles can circulate in the blood for a long period, and are extensively utilized to improve the therapeutic applications of curcumin, which is a promising strategy for more effective disease treatment [31]. Encapsulating curcumin into nanoformulations (nanocurcumin) was mainly appropriate and fruitful for the biological activity of curcumin, which enhances its bioavailability and solubility, and thereby retention in the body [32,33]. So far, many researchers showed that integrating curcumin into nanocarriers with varied approaches improves curcumin application in both in vitro and in vivo studies that involved the use of polymers and nanoparticles. Researchers developed some curcumin nanoformulations in clinical studies [34,35,36]. In another study, wounds treated by GC/L/C bilayer nanofibrous scaffold on the 7th day achieved a recovery rate about 80% compared to that the control [30]. Recent reports by [37] showed that FTIR spectra of CUR presented a characteristic broad peaks at 3496, 2923, and 1513 cm^−1^, assigned to phenolic O-H stretching vibrations, aromatic C-H stretching vibrations, and stretching vibration of benzene ring skeleton, respectively. Our result showed that the O-H stretching vibration of sodium alginate at 3371 cm^−1^ was significantly shifted to lower side. More specifically, various studies showed an observed peak of the IR spectrum of alginate at 3313 cm^−1^ (OH stretching). The FTIR of curcumin showed a peak at 3510 cm^−1^ attributed to phenolic OH vibrations [17]. In another similar study, peaks of collagen FTIR were shown as amide I (1680–1620 cm^−1^) and amide III (1300–1200 cm^−1^). Additionally, the hydroxyl band was obviously detected at 3200–3600 cm^−1^ [38].

Additionally, it is crucial to extend nanocurcumin formulations that demonstrate outstanding activities compared to microscale curcumin in wound-healing applications due to nanoparticles having a relatively larger surface area; this paramount feature of nanoparticles leads to the enrichment of the bioavailability of drugs [14,19,31]. Nanocurcumin may have higher systemic bioavailability in plasma and tissue compared to that of free curcumin [39]. Obtained results clearly showed that diabetic wounds treated by curcumin-loaded chitosan nanoparticles impregnated into collagen-alginate scaffolds have more pronounced effects on wound healing than that of a placebo scaffold [19,40]. Topical application of nanocurcumin significantly enhances the wound-healing process [40]. In recent years, magnetic nanoparticles decorated with PEGylated curcumin (MNP@ PEG-Cur) were considered due to having significantly biocompatible antitumor features [41]. Table 1 demonstrates materials that were used in wound-healing applications.

## 5. Limitations and Future Research Directions

Curcumin possess some clinical limitations that must be improved. So, researchers are concentrating on problems caused by curcumin, including poor bioavailability and solubility. Using natural polymers such as liposome, chitosan, alginate, collagen, and cellulose can reduce clinical; CUR problems. These polymers are considered due to a variety of their features, such as low cost, biocompatibility, and bioavailability.

## 6. Conclusions

We investigated the effect of nanocurcumin incorporation into an ALG/COL scaffold on wound healing in trauma diseases (Figure 7). Moreover, curcumin nanoparticles detected with an average size near 64 nm that were formed, as spherical nanocurcumin is a promising therapeutic candidate with anti-inflammatory, anticancer, and antioxidant properties. Regarding our output, the novel nano-CUR-PEGylated/COL/ALG composite could significantly accelerate wound healing by about 90% over 14 days. Curcumin nanoparticles may be more effective in improving wound healing due to unique features such as better permeability, bioavailability, and water solubility. Additionally, PEGylated nano-CUR inhibited CUR from destruction by the immune system and led to increasing its solubility.

## Figures and Tables

**Figure 1 polymers-13-04291-f001:**
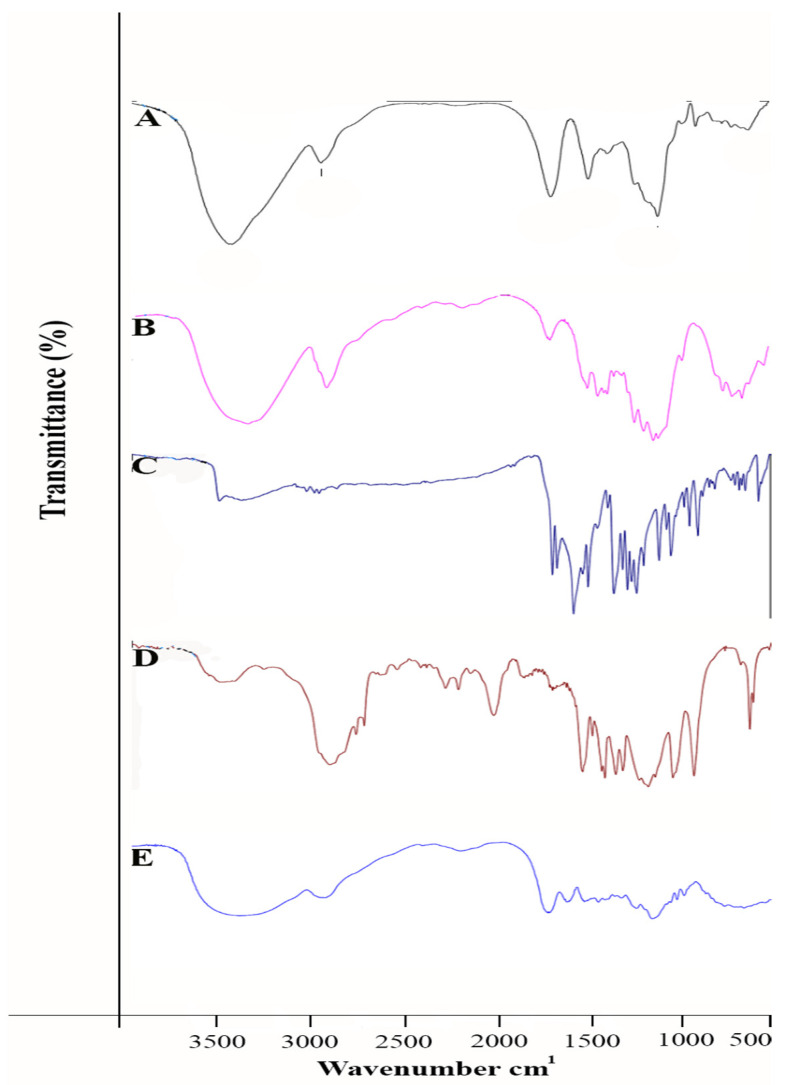
FTIR spectra of (**A**) ALG, (**B**) COL, (**C**) CUR, (**D**) PEG, (**E**) CUR, COL/ALG.

**Figure 2 polymers-13-04291-f002:**
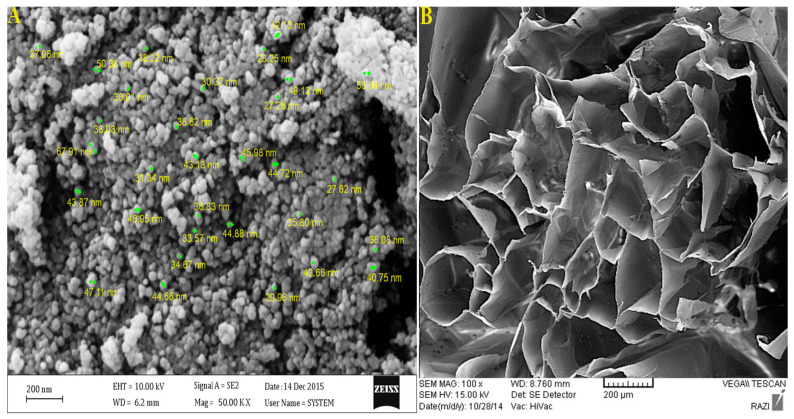
Analysis of (**A**) CUR nps and (**B**) COL/ALG composite by FE-SEM.

**Figure 3 polymers-13-04291-f003:**
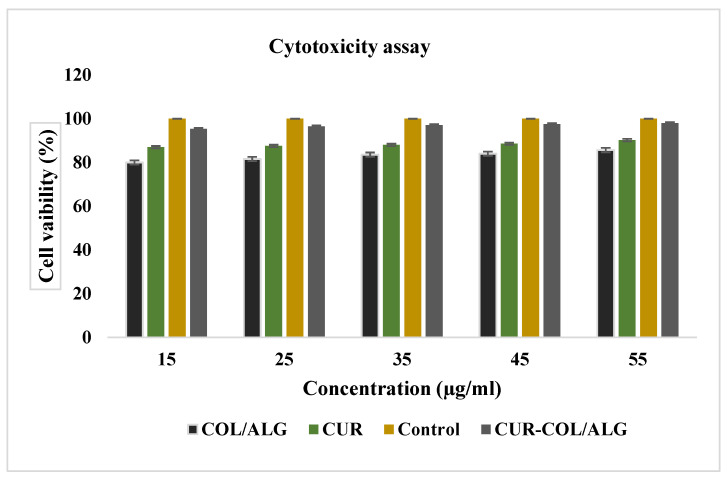
In vitro cytotoxicity of CUR, COL/ALG, CUR/COL/ALG in breast cancer cells. Cells treated with samples in different concentrations for 24 h. Percentage (%) of cell growth measured by MTT assay. Data from three independent experiments.

**Figure 4 polymers-13-04291-f004:**
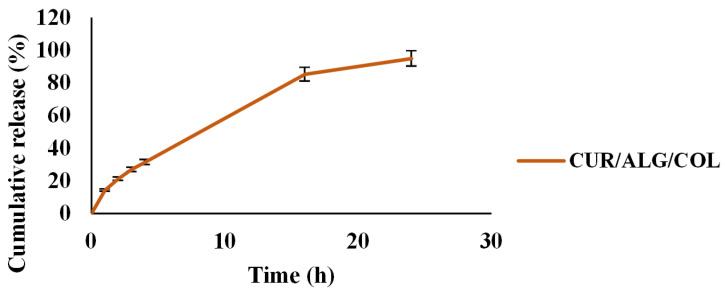
The release rate of curcumin.

**Figure 5 polymers-13-04291-f005:**
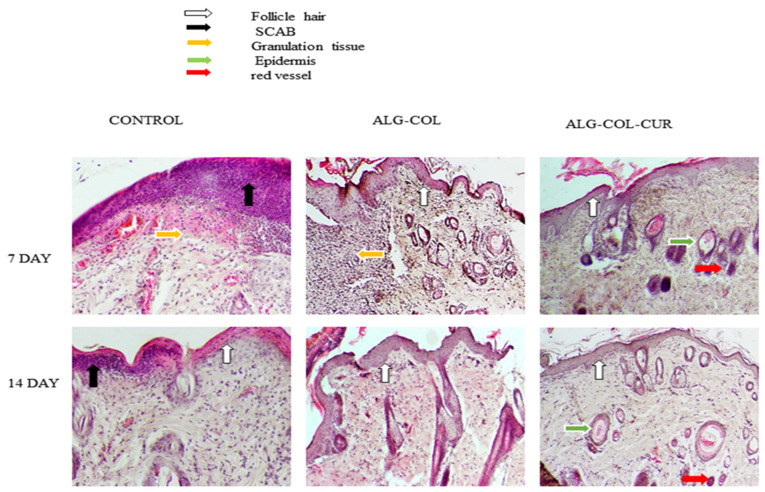
Hematoxylin and (HE) staining of skin defects treated with different treatments such as control, ALG-COL, and ALG-COL-CUR scaffolds, at 0, 7, and 14 days. HE staining at 100×.

**Figure 6 polymers-13-04291-f006:**
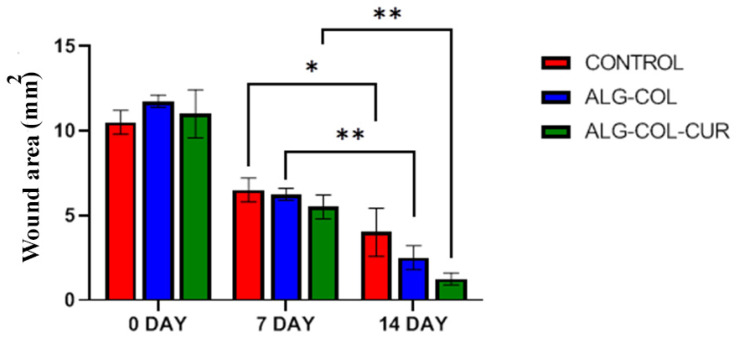
Wound sections treated with control, COL/ALG, and CUR/COL/ALG scaffolds, at 0, 7, and 14 days. Wound-area changes recorded in each group. *, ** Compared with untreated group (*p* < 0.05).

**Figure 7 polymers-13-04291-f007:**
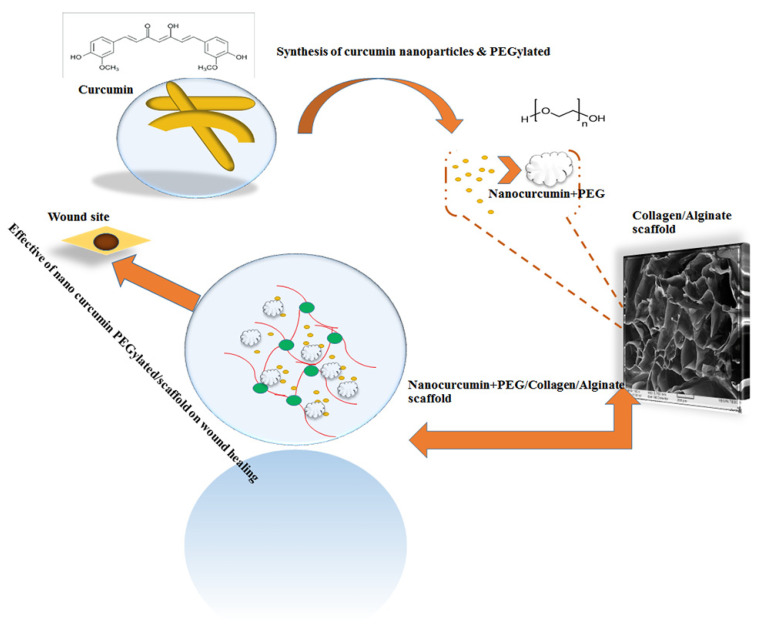
Synthesis of nanocurcumin PEGylated that was embedded in COL/ALG composite and treated wound site of skin tissue.

**Table 1 polymers-13-04291-t001:** Materials utilized in wound-healing applications.

Time	Material	Wound Healing % or *p*	Reference
9 days	Oligochitosan/curcumin/liposome	33.71	[42]
14 days	Curcumin/electrospun nanofibers	83.33	[43]
7 days	Curcumin/Lithospermi Radix Extract/*Gelatin*/*Chitosan Bilayer Nanofibrous Scaffolds*	58 ± 7	[16]
14 days	Curcumin/alginate–gelatin composite fibers	98.75 ± 4.05	[17]
15 days	Curcumin nanoparticles/collagen-chitosan scaffold	*p* < 0.01	[19]
15 days	Curcumin/chitosan nanoparticles-collagen-alginate scaffolds	98.1 ± 3.4%	[18]
10 days	curcumin/poly (lactic-co-glycolic acid) nanoparticles	75%	[30]
36 h	curcumin/gelatin-blended nanofibrous mats	40.44	[14]
12 days	Curcumin/oxidized cellulose nanofiberpolyvinyl alcohol hydrogel	80.3 ± 1.4	[24]
14 days	Curumin/hyaluronic acid	96% ± 3%	[15]

## Data Availability

The data presented in this study are available on request from the corresponding author.

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
