# Peer review of "The Effects of Curcumin Nanoparticles Incorporated into Collagen-Alginate Scaffold on Wound Healing of Skin Tissue in Trauma Patients"

_polymers, 2021, doi:10.3390/polym13244291_

Round 1

Reviewer 1 Report

I read an interesting research work entitled Effects of curcumin nanoparticles impregnated into collagen-alginate scaffold on wound healing promote of skin tissue in trauma patients. The concept of the article is interesting and suitable to publish in Polymers. This manuscript is generally well written and clearly presented however still needs to address some comments, and thus require substantial major revision to improve the quality of the manuscript.

  • A well addressed graphical scheme of study design should be inserted.  
  • Abstract should be according to Journal style and add more detail of results.
  • In the introduction section, write the novelty of the work and the problem statement clearly. More discussion about the advances about wound healing materials and what is the novelty of this research work needed. Give detailed research objectives at the end of introduction not the repetition of abstract.
  • Substantial discussion of FTIR peaks denote the values in figure and their comparison with the literature is expected during revision.
  • It is very difficult to measure the diameter of particles using SEM, sometimes not correctly. If possible authors can perform TEM analysis for the same.
  • This manuscript lacked substantial discussion of results with the recent literature authors should concentrate on this during revision. Very few papers have been referred to within the year of 2018-2020 and recent studies related to the research work.
  • Have authors checked the stability of produced materials?
  • Add one Comparative table of other materials used for wound healing applications with your research outputs.
  • Techno Economic challenges of the developed system need to be addressed. What are the limitations and future research directions that need to be described by adding a new section before the conclusions section?
  • The conclusion of the study needs to be added with the specific output obtained from the study, it could be modified with precise outcomes with a take home message.
  • Some English and grammar mistakes are present that need to be correct to improve the quality of the manuscript.

Author Response

We have answered all revises.  

Reviewer 2 Report

In this study, the authors developed the curcumin nanoparticles impregnated collagen-alginate scaffold for wound healing application. In my opinion, the manuscript is not written well and contains many flaws contains. Hence, I could not recommend this article for publication at present form.

Some critical suggestions:

  1. The title of the manuscript is not clear and written inappropriately.
  2. The introduction of the manuscript contains many improper sentences in terms of scientific and grammar. For example, the first 3 lines and “Besides, they can use for reducing the side effects, slowing release of encapsulated drugs, delay the blood circulation, poor water solubility and low efficiency for targeting tumor” What authors want to convey from the statements?
  3. The importance of curcumin was poorly written.
  4. The last three lines of the introduction were not completely conveyed the aim of this manuscript.
  5. What were the curcumin loading and encapsulation efficiency? How curcumin was formed as the nanoparticles.
  6. Section 2.5, MTT Cell Viability Assay is not valid. This phenomenon makes us to thought whether this experiment was really done or just fabricated. Some statements “After 4 h incubation, the cell culture was centrifuged at 1500 g for 5 minutes. Then, 150 ml of media was harvested from each well. Followed the plates were dried on paper towels and added 100 ml of DMSO. Absorbance was detected at 490 nm by immune absorbent assay (ELISA)”
  7. Section 2.6. The determination of the release profile is not written. How much concentration of nanoparticles was used?
  8. Section 2.7. Wound healing and hematoxylin and eosin staining. Please write properly. “All-female rats are 6-7 weeks old and weigh 145–170”, Where is the weight unit?
  9. The results section is not described well. Authors should write their results in detail.
  10. Most of the Figure legends are inappropriate. Authors do not care about anything scientific point of view, just like that written.
  11. The language of the article was very poor.
  12. Where is the conclusion?

Author Response

We have answered all revises. 

Round 2

Reviewer 1 Report

The authors have substantially revised the manuscript according to the comments.

The present form of the manuscript can be accepted for publication.

Author Response

Dear reviewer,

Thank you very much for your consideration. 

Best,

Dr. Soltani

Reviewer 2 Report

-

Author Response

Dear Reviewer,

Thank you very much for your consideration.

Best,

Dr. Soltani